# Granick revisited: Synthesizing evolutionary and ecological evidence for the late origin of bacteriochlorophyll via ghost lineages and horizontal gene transfer

Lewis M. Ward[1]*, Patrick M. Shih[2,3,4,5]*

1 Department of Earth and Planetary Sciences, Harvard University, Cambridge, Massachusetts, United States of America, 2 Department of Plant Biology, University of California, Davis, California, United States of America, 3 Environmental Genomics and Systems Biology Division, Lawrence Berkeley National Laboratory, Berkeley, California, United States of America, 4 Feedstocks Division, Joint BioEnergy Institute, Emeryville, California, United States of America, 5 Genome Center, University of California, Davis, California, United States of America

* lewis_ward@fas.harvard.edu (LMW); pmshih@ucdavis.edu (PMS)

**Data Availability Statement:** All data used in this study are derived from publicly available databases as described in the methods. NCBI Genbank accession numbers or WGS IDs as appropriate for all genomes used in this study are listed in S2

## Abstract

Photosynthesis—both oxygenic and more ancient anoxygenic forms—has fueled the bulk of primary productivity on Earth since it first evolved more than 3.4 billion years ago. However, the early evolutionary history of photosynthesis has been challenging to interpret due to the sparse, scattered distribution of metabolic pathways associated with photosynthesis, long timescales of evolution, and poor sampling of the true environmental diversity of photosynthetic bacteria. Here, we reconsider longstanding hypotheses for the evolutionary history of phototrophy by leveraging recent advances in metagenomic sequencing and phylogenetics to analyze relationships among phototrophic organisms and components of their photosynthesis pathways, including reaction centers and individual proteins and complexes involved in the multi-step synthesis of (bacterio)-chlorophyll pigments. We demonstrate that components of the photosynthetic apparatus have undergone extensive, independent histories of horizontal gene transfer. This suggests an evolutionary mode by which modular components of phototrophy are exchanged between diverse taxa in a piecemeal process that has led to biochemical innovation. We hypothesize that the evolution of extant anoxygenic photosynthetic bacteria has been spurred by ecological competition and restricted niches following the evolution of oxygenic Cyanobacteria and the accumulation of $O_2$ in the atmosphere, leading to the relatively late evolution of bacteriochlorophyll pigments and the radiation of diverse crown group anoxygenic phototrophs. This hypothesis expands on the classic "Granick hypothesis" for the stepwise evolution of biochemical pathways, synthesizing recent expansion in our understanding of the diversity of phototrophic organisms as well as their evolving ecological context through Earth history.

Table. Phylogenies used in analyses including branch support values and accession numbers/ WGS IDs for each sequence are provided as S3– S16 Figs.

**Funding:** LMW was supported by an Agouron Institute Postdoctoral Fellowship and a Simons Foundation Postdoctoral Fellowship in Marine Microbial Ecology. The funders had no role in study design, data collection and analysis, decision to publish, or preparation of the manuscript.

**Competing interests:** The authors have declared that no competing interests exist.

## Introduction

Earth's biosphere today is incredibly productive, with >99% of the organic carbon fixed per year fueled by a single metabolism—oxygenic photosynthesis [1,2]. This metabolism is an evolutionary singularity, having evolved via serial coupling of two photosystems in ancestors of crown group oxygenic Cyanobacteria, and is uniquely capable of using sunlight to power water splitting, yielding $O_2$ as a byproduct as well as providing electrons for carbon fixation [3]. Oxygenic photosynthesis evolved from more ancient forms of anoxygenic photosynthesis, which supported the biosphere early in its history [4–8]. However, the origin and early evolution of anoxygenic photosynthesis is not well understood due to the antiquity of these events and the lack of clear signatures in the rock record [1,3]. The earliest widely accepted evidence for anoxygenic photosynthesis in the rock record is in the form of depth-dependent organic carbon production in preserved microbial mats from ~3.4 Ga [9,10], and it is clear that both forms of reaction center had evolved in time to be brought back together in stem group Cyanobacteria to invent oxygenic photosynthesis and trigger the Great Oxygenation Event ~2.3 Ga [11]. However, this leaves over a billion years in which photosynthesis was likely present on Earth and actively evolving, and it is unclear how long ago the reaction centers diverged or how long the stem lineage persisted. While molecular clocks and other studies of evolution in deep time often assume that crown group synapomorphies are acquired either at the base of the crown group [12,13] or the base of the total group [14], there is no *a priori* way of determining where along a stem lineage these traits were acquired. It is therefore essential to recognize uncertainty that is inherited with long stem lineages and to make use of all available basal lineages and sister groups in phylogenetic analyses to break up long branches and reduce uncertainty in timing of acquisition of important traits [15].

While simple forms of photoheterotrophy can be supported by ion-pumping rhodopsins, there are no known examples of this metabolism driving carbon fixation [16]; true photosynthesis is only known to be supported by a more complicated pathway involving multiple components including an electron transport chain (including Complex III or Alternative Complex III), phototrophic reaction centers or photosystems, synthesis and modification of chlorophyllide pigments for light harvesting, and optionally carbon fixation to enable photoautotrophy. The capacity for reaction center-based phototrophy is scattered across the tree of Bacteria, with a handful of phototrophic lineages separated by many nonphototrophic groups (Fig 1). While early hypotheses for the evolutionary history of photosynthesis invoked vertical inheritance and extensive loss in most lineages (e.g. [17]), more recent work is more consistent with this distribution being driven by horizontal gene transfer (HGT) of phototrophy [18–21].

While previous attempts to understand the history of HGT of phototrophy have investigated relatively recent transfer events of complete phototrophy pathways [18–21,24,25], it is possible that individual components of the pathway may be transferred independently of one another, with modern phototrophs encoding chimeric pathways with components acquired from multiple sources via multiple HGT events. An example of this is indications that the (bacterio)chlorophyll synthesis pathway may have a distinct evolutionary history from the phototrophic reaction center, such as HGT of (bacterio)chlorophyll synthesis in ancestors of phototrophic Chloroflexi, Chlorobi, and WPS2 (Eremiobacterota) [21,26,27].

The extant distribution of chlorophyll versus bacteriochlorophyll synthesis is an additional puzzle. Bacteriochlorophyll is biochemically more complex to synthesize and uses lower quality light [28] yet is found today in bacteria using more ancient anoxygenic phototrophy (Table 1). Oxygenic Cyanobacteria, in contrast, use chlorophyll, which harvests higher quality light and is biochemically more straightforward to synthesize. This has led to hypotheses for the evolution of (bacterio)chlorophylls ranging from the stepwise evolution of progressively more complex

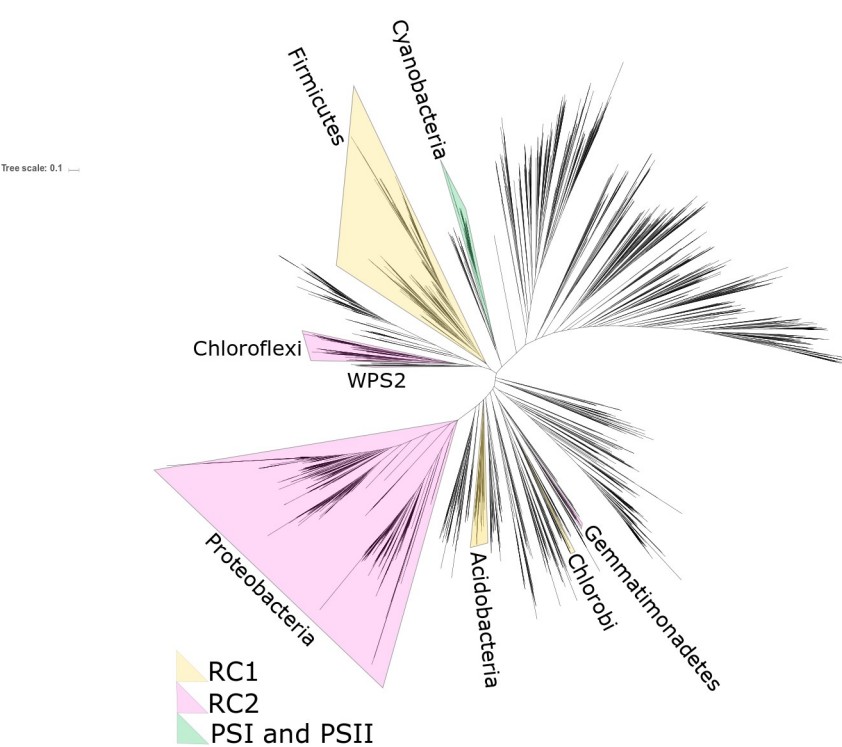

**Fig 1. Unrooted phylogeny of bacteria built with concatenated ribosomal protein sequences, following [22] with additional data from newly described phototrophs in the phylum Eremiobacterota (here labeled by their alternate name WPS2) [21,23] and the Chloroflexi class Anaerolineae [19].** Occurrences of reaction center-based phototrophy within a phylum indicated by shading of the entire phylum, color coded by reaction center type (RC1 for Type 1 anoxygenic reactions centers, RC2 for Type 2 anoxygenic reaction centers, PSI for photosystem 1, PSII for photosystem 2). For clarity, the entire phylum is highlighted even when phototrophy is restricted to only some members (e.g. the phototrophic Heliobacteria within the much broader, predominantly nonphototrophic, Firmicutes phylum).

pathways, with chlorophyll being more evolutionarily ancient than bacteriochlorophyll (i.e. the Granick hypothesis, [29]), or hypotheses invoking the complete bacteriochlorophyll synthesis pathway in the last common ancestor of extant phototrophs followed by secondary simplification of the pathway in Cyanobacteria [30]. The synthesis of bacteriochlorophyll and chlorophyll

**Table 1. Traits of extant phototroph lineages.**

|  | Phylum | RC | Pigments | C fixation | Aerobic? | Distribution of phototrophy |
|---|---|---|---|---|---|---|
| Oxyphotobacteria | Cyanobacteria | PSI, PSII | Chl A | CBB | Yes | Monophyletic, synapomorphic |
| Chloroflexia | Chloroflexi | RC2 | Bchl A, or Bchl C and Bchl A | 3HP or CBB | Yes | Monophyletic, derived |
| Anaerolineae | Chloroflexi | RC2 | Bchl A | None | Yes (predicted) | Polyphyletic, derived |
| Chloracidobacteria | Acidobacteria | RC1 | Bchl A and Bchl C | None | Micro | Monophyletic, derived |
| Heliobacteria | Firmicutes | RC1 | Bchl G | None | No | Monophyletic, derived |
| Chlorobi | Chlorobi | RC1 | Bchl A, plus Bchl C, D, E, or F | rTCA or none | No (Yes for one lineage) | Monophyletic, synapomorphic |
| Alphaproteobacteria | Proteobacteria | RC2 | Bchl A | CBB or none | Yes (some exceptions) | Polyphyletic, derived |
| Gammaproteobacteria | Proteobacteria | RC2 | Bchl A | CBB or none | Yes (some exceptions) | Polyphyletic, derived |
| Baltobacterales | Eremiobacterota (WPS-2) | RC2 | BchlA | CBB or none | Yes (predicted) | Polyphyletic, derived |
| Gemmatimonas | Gemmatimonadetes | RC2 | Bchl A | none | Yes | Monophyletic, derived |

both proceeds through chlorophyllide a, and so the two pathways share a "backbone" of shared steps, while bacteriochlorophyll synthesis has additional later steps, including those performed by the BchXYZ complex, which is homologous to the BchLNB complex involved in the synthesis of chlorophyllide a from protochlorophyllide a (S1 Table, S1 Fig). This homology provides an opportunity for querying the relative evolutionary histories of bacteriochlorophyll and chlorophyll: if the two complexes have congruent phylogenies with the exception of the absence of BchXYZ in Cyanobacteria, this would argue for a secondary simplification of the pathway as suggested by [30]. However, incongruent phylogenies would indicate an independent history of HGT, such as might be expected if bacteriochlorophyll synthesis was secondarily acquired in anoxygenic phototrophs relatively later in their evolutionary history after the divergence of crown group lineages.

Here, we perform phylogenetic analyses of various components of the phototrophy pathway, including individual steps in (bacterio)chlorophyll synthesis, and demonstrate that phylogenetic trees of these components have incongruent topologies, reflecting independent histories of horizontal gene transfer. This suggests a mix-and-match style of modular evolution by which bacteria acquire separate components of phototrophy from separate sources at different times in their evolutionary histories. Moreover, we provide support for an early diverging "ghost lineage" of extinct or undiscovered phototrophs responsible for evolutionary divergence of RC2 and BchXYZ. The genes encoding these complexes were later transferred into crown group phototrophs, leading to evolutionary novelty such as coupled photosystems and bacteriochlorophyll synthesis. We then propose an ecological model for the evolution of phototrophy in deep time, whereby competition with Cyanobacteria and environmental partitioning by $O_2$ tolerance has led to innovation, diversification, and specialization by crown group anoxygenic phototrophs, leading to diverse bacteriochlorophyll-synthesizing anoxygenic phototrophs today that radiated after the GOE, supplanting ecologically and genetically distinct chlorophyll-synthesizing anoxygenic phototrophs which fueled primary productivity in Archean time.

## Results and discussion

### Phylogenetic analyses of reaction center and (bacterio)chlorophyll synthesis proteins

We have constructed phylogenies of essential proteins for phototrophy including reaction centers and (bacterio)chlorophyll synthesis (listed in S1 Table) using representatives from all known clades of phototrophic bacteria (Table 1). Consensus trees for phototrophy proteins are shown in Fig 2, with individual phylogenies available as Supplemental Information (S2–S12 Figs).

While these trees have difficulty recovering deep evolutionary relationships due to long evolutionary distances relative to protein lengths, a problem previous recognized for interpreting the evolutionary history of these proteins (e.g. [3,26,31]), they robustly recover overall topologies and sister-group relationships that differ between different proteins. Breaking up long branches by adding newly discovered phototrophic taxa (e.g. Chloracidobacteria and Baltobacterales) improves resolution of deep branches relative to previous attempts with more limited datasets (e.g. [30]). The topologies of phylogenetic trees constructed here are largely incongruent (e.g. organismal tree, RC tree, backbone chlorophyll synthesis, and bacteriochlorophyll synthesis, Fig 2), indicating that components of the phototrophic apparatus have undergone independent horizontal gene transfer events over the course of their evolutionary history [32].

While branch support within diverse, shallow radiations such as the Cyanobacteria and the Proteobacteria were consistently poor (e.g. S3–S16 Figs), and well-supported roots were not

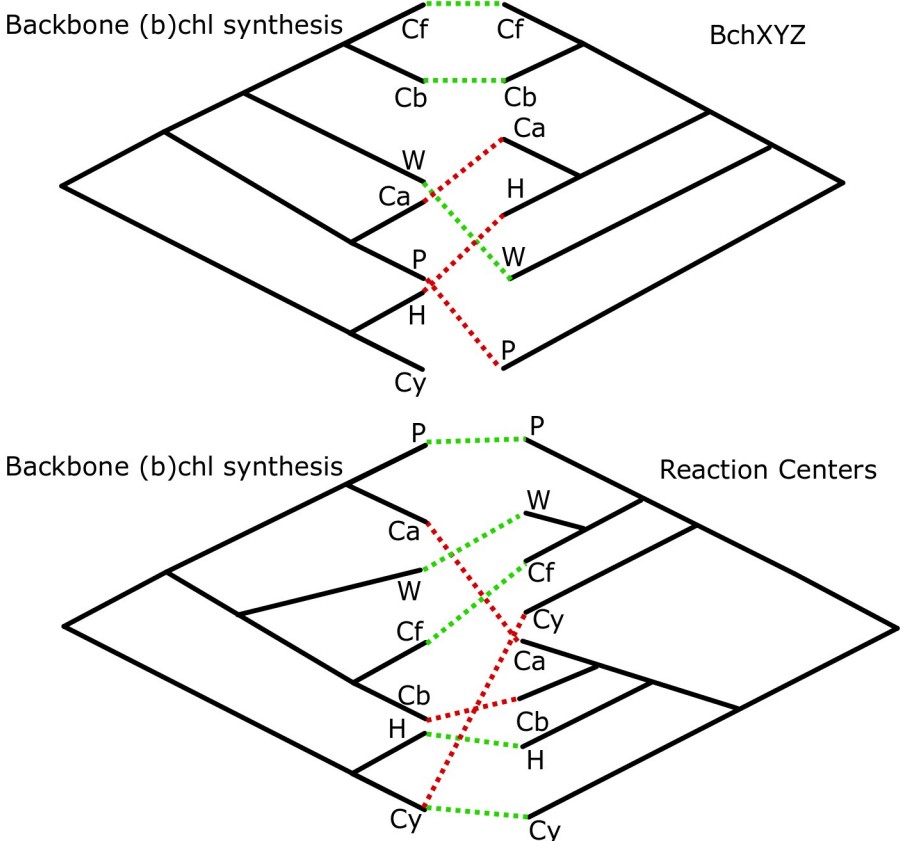

**Fig 2. Tanglegrams demonstrating phylogenetic incongruities between separate components of phototrophy pathways, including reaction centers, backbone (bacterio)chlorophyll synthesis, and the BchXYZ complex involved in bacteriochlorophyll synthesis; red lines indicate phyla with distinct branching order in phylogenies on the left and right, likely reflecting independent histories of horizontal gene transfer.** Taxon abbreviations: Cf, Chloroflexi; Cb, Chlorobi; W, WPS2/Eremiobacterota; Ca, Chloracidobacteria; H, Heliobacteria; P, Proteobacteria (including Gemmatimonadetes); Cy, Cyanobacteria.

recovered for all proteins investigated (e.g. long branches between BchL/N/B and BchX/Y/Z made root placement inconsistent, S3–S16 Figs, the consistent topology between consensus BchLNB and BchHDIM trees clustered at higher taxonomic levels (i.e. order through phylum) and robust rooting of BchH and BchM via characterized homolog outgroups allows extrapolation of branching order from these trees to the consensus backbone (bacterio)chlorophyll synthesis tree. While uncertainty in early branching order may affect hypotheses of the directionality of HGT invoked in hypotheses for evolutionary history of phototrophy, major sister-group relationships and incongruent topologies between different complexes are robustly recovered, supporting overall evolutionary trends even if the exact number, timing, and directionality of HGT events are uncertain. As a result, interpretations of the role of HGT in driving the evolution of phototrophy are considered robust here, even if the scenario depicted in Fig 3 is only a hypothesis.

Backbone (bacterio)chlorophyll synthesis genes have a somewhat different evolutionary history than downstream bacteriochlorophyll-specific genes, which have a distinct evolutionary history from phototrophic reaction centers, which have independent histories from carbon fixation pathways—all of which are incongruent with organismal phylogenies. Taken all

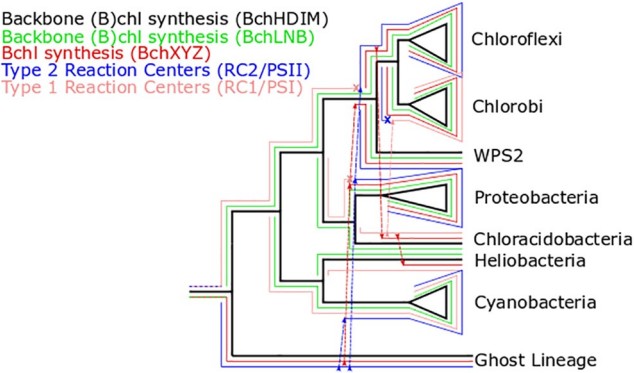

**Fig 3. Cartoon overlayed phylogenies to demonstrate hypothesized history of HGT and ghost lineage.** Underlying topology derived from backbone (bacterio)chlorophyll synthesis genes (BchH/D/I/M) (black). BchLNB and BchXYZ are derived from a common ancestor in stem group phototrophs. BchLNB (green) was inherited together with BchHDIM into extant phototrophs; BchXYZ (red) diverged in the ghost lineage before being introduced into extant anoxygenic phototrophs via HGT (a first HGT event introduced it into the stem of the proteobacterial lineage; a second HGT event introduced it into the stem of the WPS2/Chlorobi/Chloroflexi lineage; subsequent HGT introduced it into the Chloracidobacteria and Heliobacteria lineages). Type 1 reaction centers (RC1 and PSI) and Type 2 reaction centers (RC2 and PSII) diverged in stem lineage phototrophs. Type 1 reaction centers (peach) were vertically inherited into extant phototrophic lineages. Type 2 reaction centers (blue) diverged in the same ghost lineage as BchXYZ, and were introduced into extant clades via HGT (first into stem group Cyanobacteria, leading to PSII, then into stem group Proteobacteria and the stem lineage of WPS2, Chloroflexi, and Chlorobi). The most parsimonious history consistent with the data involves a secondary replacement of RC2 with RC1 in Chlorobi. Alternative evolutionary histories are similarly parsimonious, but all involve many events of HGT of individual phototrophy components and most involve secondary loss and replacement in some lineages. The inclusion of one or more ghost lineages improves parsimony and provides a good explanation for long branches between RC1/RC2 and BchLNB/BchXYZ homolog pairs. A summary of hypothesized HGT events is presented in Table 2.

together, the reticulated and piecemeal nature of this incongruencies suggest that the evolution of phototrophy, like many other metabolic pathways, is modular [3,15,19,33,75,76].

## Evolutionary history of photosynthesis-associated proteins

The incongruent topologies of trees built with various phototrophy-related proteins (Fig 2) suggests that these proteins may not share a unified evolutionary history but may instead have undergone independent horizontal gene transfer events. By overlaying components of the phototrophy apparatus onto the backbone tree made from the consensus topology of proteins involved in shared early steps of (bacterio)chlorophyll synthesis, we can reconstruct a hypothetical evolutionary history of the genes, even if a largely unconstrained cooccurring history of organismal transfer must also be occurring throughout stem lineages (Fig 3). For example,

**Table 2. Hypothesized HGT events of phototrophy modules.**

| Enzymes | Donor Organism | Recipient Organism | Notes |
|---|---|---|---|
| RC2 | Ghost Lineage | Proteobacteria | |
| RC2/PSII | Ghost Lineage | Cyanobacteria | Followed by change of function from RC2 to PSII |
| BchXYZ | Ghost Lineage | Proteobacteria | Followed by change of function |
| BchXYZ | Proteobacteria | Chloroflexi+Chlorobi+WPS2 | |
| RC2 | Proteobacteria | Chloroflexi+Chlorobi+WPS2 | Associated with loss of RC1 |
| BchXYZ | Chloroflexi+Chlorobi | Chloracidobacteria | |
| RC1 | Chloracidobacteria | Chlorobi | Associated with loss of RC2 in Chlorobi |
| BchXYZ | Chloracidobacteria | Heliobacteria | |

the node representing the last common ancestor of (bacterio)chlorophyll synthesis genes in Chlorobi and Chloroflexi almost certainly did not occur in crown group members of either of those phyla or their last common ancestor, but instead independent HGT events introduced phototrophy to those groups following the divergence of their (bacterio)chlorophyll pathways in other host lineages. In some cases, we see transfer of the complete phototrophy pathway (e.g. into Gemmatimonadetes from Proteobacteria, [18]), while in others it appears that some components were transferred independently of others (e.g. bacteriochlorophyll synthesis and reaction centers into Chlorobi and Chloroflexi, respectively). In many cases, we see transfer of phototrophy but not carbon fixation (e.g. Anaerolineae, Gemmatimonadetes, [18,19,34]) though we also see cases of gain, loss, and reacquisition of carbon fixation (e.g. within the Chloroflexia, [15]).

Below, we describe a hypothesis for the evolutionary history of phototrophy genes that is consistent with all of the available data, from the "ur-phototroph" (with the prefix "ur-"denoting the first or proto-version of a thing, here referencing the first phototrophic lineage to evolve, i.e. the first member of stem and total group phototrophs) until the radiation of extant clades of phototrophs (the crown group). The history presented here relies on undiscovered, likely extinct, "ghost lineages" which served as the source of genes to crown group phototrophs during ancient transfer events. The concept of "ghost lineages" is carried over from traditional fossil record-based phylogenetics, where it refers to a lineage which is inferred to exist but which has no known fossil record [35,36]. Importantly, this history is considered from the perspective of phototrophy genes, not of the organisms which harbor them.

The ur-phototroph likely utilized a single ancestral reaction center (almost certainly a homodimer that formed a stem lineage prior to the RC1/RC2 divergence, and which may have been biochemically more similar to heliobacterial RC1 than other extant types, made up of a relatively simple and inefficient homodimer which loosely bound mobile quinones to drive cyclic electron flow, which was adapted in the various reaction center and photosystem lineages to optimize reactions and eventually to adapt to oxygen, [37]) and chlorophyll a, synthesized using a DPOR complex ancestral to both BchLNB and BchXYZ. Eventually, two lineages of phototrophs diverged, either due to speciation of a single organismal lineage or due to HGT of phototrophy genes into a second host organism. One lineage (the "ghost lineage") possessed a reaction center that evolved into RC2 and a BchLNB-like complex that eventually evolved into BchXYZ but which functioned as a DPOR complex. The other lineage possessed an ancestral RC1 and BchLNB in order to synthesize chlorophyll. The RC1 lineage diversified, with RC1 and chlorophyll synthesis in several lineages. Eventually, HGT of RC2 from the ghost lineage into stem group Cyanobacteria led to the evolution of oxygenic photosynthesis. This triggered ecological restructuring and widespread evolutionary adaptation, including further HGT from the ghost lineage into other lineages of anoxygenic phototrophs. This included HGT of the BchXYZ complex from the ghost lineage, perhaps into stem group Proteobacteria, leading to the coupling of BchLNB and BchXYZ in series to lead to bacteriochlorophyll synthesis and the ability of anoxygenic phototrophs to better compete in deeper, lower oxygen regions of microbial mats and water columns (see below section "Ecological perspectives on the evolution of photosynthesis"). Further HGT of RC2 from the ghost lineage into other anoxygenic phototroph lineages led to further adaptation and specialization of anoxygenic phototroph lineages to specialized environments. The long branches between homologous proteins in the BchLNB and BchXYZ complexes is consistent with their early divergence sometime during Archean time before the radiation of crown group phototrophs; the congruence between BchLNB and other backbone chlorophyll synthesis genes suggests that these genes have largely been inherited together into extant phototrophs, while the differing topology of BchXYZ indicates an

independent history of HGT (S2 Fig), in this scenario driven by HGT of BchXYZ from the ghost lineage into crown group anoxygenic phototrophic clades, allowing them to produce bacteriochlorophyll.

For simplicity one ghost lineage is described here, but it is likely there were many lineages that are not represented in characterized phototrophs today, both diverging from the stem of the phototroph tree as well as from throughout the crown group. For example, the ghost lineage invoked here is hypothesized to use a type 2 reaction center (though this may have been a more ancient homodimeric form, as heterodimerization postdated the divergence of RC2 and PSII, [38]), but there are also suggestions from the carbon isotope record suggesting ancient Wood-Ljungdahl utilizing Type 1 phototrophs [1], indicating that there are multiple undiscovered or extinct phototroph lineages.

Some phototrophic Chloroflexi in the Anaerolineae class lack BchLNB as well as genes for the alternative light-activated POR complex [19,34,39] despite evidence for fluorescence microscopy-based evidence for functional bacteriochlorophyll a synthesis in at least some of these organisms [40]. This appears to be a derived trait based on their placement in BchHDI and BchXYZ trees. The genomes of these organisms are derived from metagenomic data and so are somewhat incomplete, but the probability that the organisms contain these genes but they weren't recovered in the MAG is incredibly low (estimated by MetaPOAP as ~$10^{-12}$, [41]). Instead, this appears to be a case of secondary loss potentially coupled with bifunctionalization of BchXYZ to perform the reduction of both the C17/C18 double bond normally reduced by BchLNB as well as the C7/C8 double bond normally reduced by this complex. This seems feasible, as chimeras of other homologs of these genes have been demonstrated to be functionally exchangeable (e.g., [42,43]). Notably, Cheng et al. demonstrated the enzymatic flexibility of the complex with ChlL functionally replacing the NifH subunit in the more distant homologous nitrogenase complex (NifHDK) [42]. Following the evolutionary history proposed in Fig 3, the BchXYZ complex is descended from a BchLNB-like complex for the reduction of the C17/C18 double bond that was later coopted to instead reduce the C7/C8 double bond to enable bacteriochlorophyll synthesis. However, isolation of phototrophic Anaerolineae and biochemical characterization of their BchXYZ complexes will be necessary to test this hypothesis.

## Ecological perspectives on the evolution of photosynthesis

Overlaid onto the history of HGT-driven evolution of phototrophy is also a history of adaptation to changing environmental conditions, particularly the rise of atmospheric oxygen. The earliest evidence for photosynthesis on Earth is found in rocks of Paleoarchean age [9,10], but many of the extant groups of anoxygenic phototrophs are thought to represent relatively recent radiations (i.e. post-GOE) [3,15,19,21]. Thus, many modern anoxygenic phototrophs are therefore not relicts of the Archean Earth, but instead have undergone billions of years of evolution, leading to extant phototrophs that are a palimpsest of genetic innovation and horizontal gene transfer, with over a billion years of evolution hidden in stem lineages. Straightforward comparative genomic and phylogenetic techniques extrapolate only from extant diversity and therefore may overlook nuance and complexity in the early evolution of pathways before the emergence of the last common ancestors of extant clades. A complementary alternative approach is to develop theories for the ecological drivers of the evolution of phototrophy, which can produce hypotheses which are testable with the biological record, and which can be used to choose between competing hypotheses which are equally supported by the biological record but of which some may be more ecologically viable.

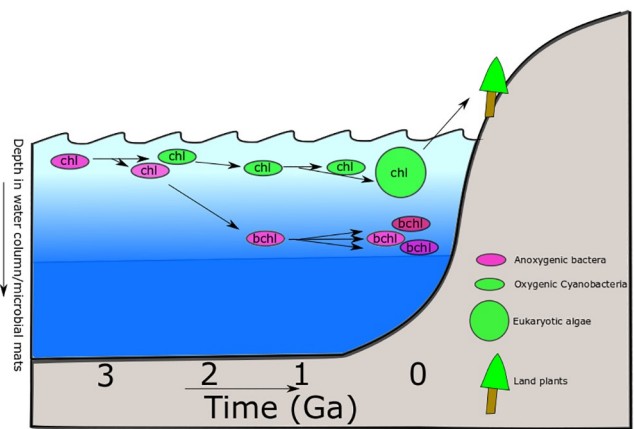

**Fig 4. Cartoon timeline of phototroph evolution as hypothesized here.** The "ancestral" phototroph was anoxygenic and utilized chlorophyll pigments, adapted to high light. At least two lineages of phototrophs diverged during Archean time, giving rise to the ancestors of Type 1 Reaction Centers and the BchLNB complex, and the Type 2 Reaction Centers and the BchXYZ complex. Eventually, HGT of a Type 2 RC into an RC1 and BchLNB-containing proto-cyanobacterium enabled the evolution of oxygenic photosynthesis, still using chlorophyll pigments. As oxygenated surface waters and competition with Cyanobacteria forced anoxygenic phototrophs into lower light regions of the water column or microbial mats (where oxygen is lower and electron donors are more abundant), they underwent adaptation to lower quality light. This included the HGT of BchXYZ complexes into BchLNB-containing lineages, allowing the innovation of bacteriochlorophyll pigments, probably first with bchl a. Eventually, anoxygenic phototrophs diversified in terms of organisms (including extant groups), pigments (bchl c-g), and reaction centers (further HGT of RC2). Oxygenic phototrophy diversified via eukaryotic endosymbiosis (primary and higher order) and colonization of land by plants. While the relative timing of these events can be inferred from comparative biology, absolute timing of many of these events is only poorly constrained if at all, based on the sparse microfossil and biomarker record of early phototrophic microbes and molecular clock estimates for the antiquity of crown group clades.

While in principle the innovation and expansion of bacteriochlorophyll synthesis could have occurred before the evolution of oxygenic photosynthesis in Cyanobacteria, we propose that it was instead the expansion of oxygenic phototrophs and the rise of oxygen that likely led to the rampant HGT of the capacity for bacteriochlorophyll synthesis and overall success of this strategy among taxonomically and ecologically diverse anoxygenic phototrophs (Fig 4).

Before the rise of oxygen ~2.3 Gya, surface environments were anoxic [44–49], and electron donors for anoxygenic photosynthesis would have been reasonably abundant [4–6,50]. As a result, these environments would have been permissive to anoxygenic phototrophs, allowing them to thrive in shallow water environments and as the major primary producers in microbial mats [1]. At this time, anoxygenic phototrophs would have been free to exploit abundant, high-quality light conditions to which chlorophyll a is well adapted. The earliest anoxygenic phototrophs therefore would have had little to no evolutionary pressure to evolve more biochemically complex bacteriochlorophyll pigments. Anoxygenic phototrophs may have specialized to particular environments where electron donor compounds or other nutrients were especially abundant (e.g. volcanic environments or near shallow hydrothermal vents), but otherwise would have had little barrier to dispersal and colonization of widespread environments.

By 2.3 Gya, early Cyanobacteria had evolved oxygenic photosynthesis, leading to the accumulation of atmospheric $O_2$ and a sharp increase in primary productivity associated with this metabolism [1,6,8,11,51–53]. As Cyanobacteria are quite oxygen tolerant relative to other phototrophs and are not dependent on limited electron donors, these organisms would have been capable of colonizing all shallow aquatic environments, displacing anoxygenic phototrophic communities and leading to a distribution of phototroph types more similar to that seen today. Oxygenic phototrophs are able to grow in well-oxygenated surface environments

with largely unobstructed sunlight, and so have had little evolutionary pressure to evolve away from using ancestral chlorophyll pigments. Anoxygenic phototrophs, in contrast, would have been unable to compete with Cyanobacteria in surface environments where they would experience oxygen toxicity as well as limited availability of electron donors due to biological or abiotic oxidation driven by $O_2$. As a result, anoxygenic phototrophs would be restricted to environments deeper in water columns or microbial mats where oxygen concentrations are lower and electron donors are more available. As these environments typically underlie cyanobacterial populations (e.g., microbial mats), anoxygenic phototrophs are light-limited in the wavelengths absorbed by cyanobacterial pigments—including chlorophyll (e.g. [54]). These organisms therefore have undergone significant evolutionary pressure to evolve light-harvesting pigments that are shifted to different wavelengths that reach these deep layers. Bacteriochlorophylls fill this role well: bacteriochlorophyll pigments have peak absorbances at longer wavelengths than chlorophyll, resulting in less energy absorbed per photon but allowing them to work deeper in the water column/microbial mat, once more light has been absorbed (both by the medium and by Cyanobacteria with chlorophylls) [55,56]. This scenario is consistent with the ancestral use of chlorophyll in anoxygenic lineages followed by multiple HGT-enabled acquisitions of BchXYZ and therefore bacteriochlorophyll synthesis in anoxygenic lineages after the GOE. Ecological exclusion of anoxygenic phototrophs from high-light surface environments by competition with Cyanobacteria and the distribution of $O_2$ and electron donors likely therefore provided the evolutionary pressure for the initial evolution of bacteriochlorophyll pigments as well as to drive horizontal gene transfer leading to the widespread adoption of bacteriochlorophyll pigments by diverse anoxygenic phototrophs. Eventually, the plastid endosymbiosis event would give rise to a wide diversity of eukaryotic oxygenic phototrophs [57–59]; thus, the niche adaptation and competition with anoxygenic phototrophs would later be expanded beyond Cyanobacteria to include algae, and ultimately the rise of plants. Although the exact timing of the plastid endosymbiosis is still widely debated, the majority of studies agree that it is of Proterozoic origin, after the radiation of Cyanobacteria [60–62]. Compared to anoxygenic phototrophic lineages, the chlorophyll requirements/composition of Cyanobacteria, algae, and plants are much more similar (i.e., Chl a and b)–albeit with some notable variations [63–65]–which is reflective of the common evolutionary history of these three major groups.

As anoxygenic phototrophic lineages are typically restricted to particular niche environments, island biogeography-like evolutionary radiations have likely led to the diversification of anoxygenic phototrophs seen today, including the diversification of their bacteriochlorophyll pigments. Today, oxygenic phototrophs (Cyanobacteria, algae, and plants) are essentially ubiquitous in habitable Earth surface environments. Cyanobacteria therefore have a largely cosmopolitan distribution, with related species found in diverse environments around the world. However, anoxygenic phototrophs are typically restricted to more limited environments such as stratified water columns [66], hot springs (e.g. [39,67–69]), and geothermal soils (e.g. [70]) where the presence of sulfide and/or high temperature inhibit Cyanobacteria, though a few lineages of phototrophic Proteobacteria and Eremiobacteria are adapted to more widespread photosynthetic or photoheterotrophic niches in association with plants (e.g. [21,71]). These niches are highly localized, discrete environments with specialized geochemical conditions, limiting the dispersal of anoxygenic phototrophs between sites and inhibiting the ability of anoxygenic lineages to colonize new environments.

Further evolution has of course occurred over the billions of years since the rise of Cyanobacteria, including further HGT (e.g. the transfer of many genes, including BchLNB and the alpha-carboxysome from Proteobacteria into *Synechococcus* and *Prochlorococcus*, [26]), adaptation of anoxygenic phototrophs to their preferred niches (including diversification of

bacteriochlorophylls), and the substitution of ancestral oxygen-sensitive enzymes with $O_2$-tolerant versions, particularly in the Cyanobacteria and Proteobacteria. For example, the innovation of the POR enzyme complex as an alternative to the oxygen-sensitive BchLNB complex in oxygenic Cyanobacteria. DPOR (the BchLNB complex) is oxygen sensitive, and doesn't work at $O_2$ concentrations higher than saturation under ~3% present atmospheric levels [72], most likely leading to the evolution of POR as an oxygen-tolerant alternative in stem-group Cyanobacteria prior to the radiation of the crown group ~2 Ga [11]. POR has also been occasionally acquired by Proteobacteria [73], reflecting continuing HGT of phototrophy proteins subsequent to the diversification of crown group lineages.

## Combined perspectives on the evolution of photosynthesis

The evolution of photosynthesis is modular, involving not only independent HGT of carbon fixation and Reaction Centers but also of separate components of (bacterio)chlorophyll synthesis. The relationships among characterized extant phototrophs suggests one or more "ghost lineages", some of which likely diverged prior to the radiation of the crown group. Chlorophyll synthesis appears to be more ancient than bacteriochlorophyll synthesis, consistent with the Granick hypothesis [29]. This also results in an evolutionary scenario for the origin of bacteriochlorophyll synthesis analogous to the origin of coupled photosystems for oxygenic photosynthesis, whereby divergence of paralogs followed by reintroduction into a single host organism allows biochemical innovation (e.g. [74]). The early divergence between RC1 and RC2, and between BchLNB and BchXYZ, suggested by this history is consistent with the long branches between these sets of homologs relative to divergence within each.

The modular nature of the evolution of phototrophy is similar to broader trends in the evolution of pathways including denitrification [75], methanotrophy [33,76], and high-potential metabolism in general [3]. The ability of microbes to exchange components of preexisting metabolisms and recombine them into new and innovative pathways appears to be a major driver of innovation.

The innovation of bacteriochlorophyll by coupling in series of the orthologous BchLNB/BchXYZ complexes is analogous to the innovation of coupling divergent photosystems to drive oxygenic photosynthesis. Although the early evolution of phototrophy involved extensive shuffling of phylogenetic relationships via HGT and a role for extinct stem lineages, we can understand this history by careful comparative phylogenetics as long as we have sufficient sampling of extant diversity. Coupling this understanding to analysis of the rock record, potentially supported by molecular clock analysis, may allow us to tie absolute ages to these phylogenies casting events in relative time. For example, the organic biomarker and molecular clock evidence for the evolution of phototrophy in the Chlorobi by 1.6 Ga [77], molecular clock estimates for the origin of phototrophic Chloroflexia around 1 Ga [15], and the rise of atmospheric oxygen due to total group oxygenic Cyanobacteria around 2.3 Ga can provide some constraints. The fact that each of the clades of anoxygenic phototrophs appears to have acquired bacteriochlorophyll synthesis in stem lineages before the radiation of individual crown groups indicates that the radiations of extant anoxygenic phototroph clades occurred relatively late in Earth history after the origin of oxygenic photosynthesis in Cyanobacteria. This is consistent with estimates for increasing diversity of bacterial lineages through geologic time [78]. The taxonomic affinity and phenotypes of ancient phototrophs responsible for fueling Archean productivity remain unknown, perhaps due to the extinction of these bacterial lineage following the rise of oxygen.

The ecological partitioning of anoxygenic phototrophs into localized environments with lower oxygen concentrations and lower light energy may have led to an island biogeography-

like adaptive diversification of different phototrophic lineages to distinct environments. Discrete anoxygenic phototrophic lineages have adapted to particular preferred environments, such as hot springs for Chloroflexi and Chloracidobacteria [19,68,69], thermal soils for Heliobacteria [79], stratified water columns for Chlorobi, and acidic plant-associated niches for Baltobacterales [21] (though exceptions do occur, such as Chloroflexi in carbonate tidal flats, [34,80], and Heliobacteria in hot springs, [70]). Adaptations to these specialized environments may have been driven by island biogeography-like evolutionary trends in which particular phototroph lineages have been isolated and evolved not in specific locations but in particular geochemical settings.

## Conclusions

The origin and early evolution of phototrophy is not clearly revealed by the rock record, and care must be taken in reading the biological record; in particular, the application of comparative biology to investigating the early evolution of traits in stem lineages before the last common ancestor of extant members of a clade is inherently challenging [1]. However, large-scale comparative phylogenetics of organismal relationships and phototrophy-related proteins can provide some insight into the early evolution of photosynthesis, especially given that the independent histories of genes involved with phototrophy allow some insight into the nature of stem groups and extinct lineages in instances where genes from these organisms have been horizontally transferred and then inherited into extant organisms. These data are particularly useful once coupled to consideration of the ecological context of the organisms in question, which can help to choose between otherwise equally viable hypotheses. As we've shown here, phylogenetic relationships provide abundant evidence of horizontal gene transfer of phototrophy-related proteins, though the directionality of transfers and the relationships among the deepest branches remain ambiguous in many cases. These relationships are consistent with an early evolution of chlorophyll synthesis followed by a post-GOE radiation of bacteriochlorophyll-synthesizing anoxygenic phototrophs driven by ecological competition and niche partitioning by Cyanobacteria.

Consistent with other analyses of (bacterio)chlorophyll synthesis protein phylogenies (e.g. [26,30]), we were unable to recover robust, consistent branching order of deep divergences in many protein families (e.g. BchLNB and BchXYZ), though sister group relationships appear robust for each protein. The inconsistent branching order of deep divergences in rooted phylogenies of individual subunits of the BchLNB and BchXYZ complexes may be an artifact of long branch attraction and saturation of variable sequences in relatively small soluble proteins over billions of years of evolution, or this difference in branching order may reflect actual differences in evolutionary history, whereby subunits of these complexes underwent independent horizontal gene transfer events early in their histories. It is difficult (if not impossible) to distinguish between these possibilities given the limited sequence data and long evolutionary timescales with which we are left. Nonetheless, broad evolutionary trends and shallower sister-group relationships were robustly recovered, and these clearly indicate that different components of the complete phototrophy pathway have independent evolutionary histories.

These limitations to interpreting the early evolutionary history of (bacterio)chlorophyll synthesis reflect a larger problem that must be confronted in phylogenetic analysis over long geological timescales—sufficient information may not be left in the biological record to answer all questions (e.g. [81]). The mutational saturation rate of protein sequences, particularly for short poorly conserved soluble proteins, limits the evolutionary timescale over which sequence-based phylogenies remain meaningful. The amount of evolutionary time represented by the diversity of (bacterio)chlorophyll synthesis proteins is on the edge of the range at which

phylogenetic relationships are meaningfully recoverable, and the amount of evolutionary distance involved in outgroups necessary for rooting further complicate the recovery of meaningful evolutionary histories of these proteins (e.g. [3,31]). We therefore find particular value in integrating ecological scenarios for the evolution of phototrophy and (bacterio)chlorophyll synthesis as an independent means of supplementing the limited resolution available from extant sequence data.

This model predicts several potentially testable hypotheses, including the potential existence of a previously undiscovered early-diverging chlorophyll a-synthesizing anoxygenic phototrophs including the "ghost lineage" depicted in Fig 3 which utilizes a basal (perhaps still homodimeric) form of RC2 along with a BchXYZ-like complex to produce chlorophylls. Environmental metagenomic sequencing has proven incredibly powerful for recovering novel phototrophic lineages (e.g. [19,21,23,68]), and so it remains conceivable that a larger diversity of phototrophs may be recovered which can test the evolutionary history described here. However, it remains a strong possibility that this lineage has gone extinct, or has lost the capacity for phototrophy.

## Methods

### Summary

To compare evolutionary relationships among steps in (bacterio)chlorophyll synthesis, phylogenies were constructed for individual proteins in steps involved in the conversion of protoporphyrin IX to protochlorophyllide a (the shared "backbone" (bacterio)chlorophyll synthesis pathway shared in all reaction center-based phototrophs) as well as the subsequent conversion of protoporphyrin IX to specific chlorophyll and bacteriochlorophyll pigments found in only some phototrophs. Congruence of tree topology between phylogenies of different proteins was taken as indicative of shared evolutionary history (i.e. vertically inherited or horizontally transferred together, but not transferred individually), while incongruence was taken as an indication of independent histories of horizontal gene transfer (i.e. HGT of a subset of (b)chl synthesis proteins rather than of the entire pathway).

### Protein phylogenies

Protein translations of all bacterial genomes available from Genbank were downloaded on 3 April 2018. This database was supplemented with data from [19,23] for phototrophic Anaerolineae and Eremiobacterota, respectively. The database was queried with the BLASTP function of BLAST+ [82] using reference sequences from *Chloroflexus aurantiacus* and an e value cutoff of 1e-10. Sequences were aligned with MAFFT [83]. Alignments were curated in Jalview [84] for sequence completeness and quality of alignment. In cases where MAFFT produced poor alignments, these were reprocessed with MUSCLE [85]. Phylogenetic trees were built with RAxML [86] on the CIPRES science gateway [87]. TBE support values were calculated from RAxML bootstraps with BOOSTER [88]. Trees were visualized with the Interactive Tree of Life [89].

### Quality control

Following initial tree construction, iterative trimming of alignments was performed to remove incomplete sequences and functionally divergent homologs not involved in (bacterio)chlorophyll synthesis (e.g. nitrogenase NifD and NifH sequences returned by BchN and BchL searches, respectively). Outgroups were retained for proteins that displayed relatively short evolutionary distances and for which deep branches were robustly recovered (e.g. BchH and

CobD), but trees were left unrooted when long evolutionary distances to outgroup sequences led to inconsistent rooting and deep branching order (e.g. between BchL and BchX). Congruent topologies between rooted and unrooted trees in related steps in (bacterio)chlorophyll synthesis (e.g. BchH with BchI and BchD) was considered sufficient to produce rooted consensus trees for steps in the (bacterio)chlorophyll synthesis pathway (i.e. insertion of Mg into protoporphyrin IX for BchH/D/I, the first committed step in (bacterio)chlorophyll synthesis, shared between all reaction center-based phototrophs).

For phototroph lineages only characterized via incomplete metagenome-assembled genomes (e.g. Eremiobacterota, Anaerolineae), the likelihood that missing genes may be present in the source genome was estimated with the False Negative estimate function of Meta-POAP [41].

For steps in (bacterio)chlorophyll synthesis catalyzed by proteins that are more promiscuous and can be recruited into and from distinct pathways (e.g. methyltransferases), and for steps that can be catalyzed by multiple poorly characterized proteins (e.g. C-8 vinyl reductase, [90]), we do not report phylogenies as these were deemed unreliable for recording robust deep evolutionary relationships.

Due to very low support values for deep nodes and extensive artifacts, concatenated protein trees (e.g. BchLNB) were deemed unreliable, likely related to extensive, independent HGT of individual subunits (particularly within the Proteobacteria, e.g. [20]).

## Organismal phylogeny

Unrooted concatenated ribosomal protein trees of the bacterial domain were constructed following methods from [22], with eukaryotes and archaea omitted and phototrophic members of Anaerolineae and Eremiobacterota added [19,23,33]. Following addition of new sequences, sequences were realigned with MAFFT [83] and the tree calculated with RAxML [86] on the CIPRES science gateway [87] and visualized with iToL (Letunic and Bork 2016).

## Supporting information

**S1 Fig. Simplified diagram of chlorophyll a and bacteriochlorophyll a synthesis from protoporphyrin IX.** Steps leading to chlorophyllide a synthesis are shared by all photosynthetic bacteria. From chlorophyllide a, a single enzymatic step can produce chlorophyll a (right branch) as performed in Cyanobacteria, or multiple steps can be taken to produce bacteriochlorophyll a (left branch) as performed in most characterized anoxygenic phototrophs. While some steps can be performed by multiple enzymes and some enzymes may act on multiple substrates, allowing some steps to be performed in different orders in different organisms or in parallel in a single organism, the (b)chl biosynthesis pathway is depicted here as a simplified linear pathway for the sake of clarity. Not shown are branch points leading to more evolutionarily derived alternative pigments including bchl c, bchl g, chl b, and chl d.
(PDF)

**S2 Fig.** A) Consensus phylogenies of steps in (bacterio)chlorophyll synthesis. Major radiations are collapsed at the phylum level. The Chloroflexi clade includes phototrophic Chloroflexia as well as multiple lineages of phototrophic Anaerolineae. The Proteobacteria clade includes Gemmatimonadetes and in some cases (BchL/N/B) the Synechococcus/Prochlorococcus clade of Cyanobacteria. Consensus tree for earliest dedicated steps in (bacterio)chlorophyll synthesis, including BchH, BchI, and BchD for conversion of protoporphyrin IX to Mg-protoporphyrin IX, and BchM for further conversion to Mg-protoporphyrin monomethyl ester. Consensus tree is rooted based on consistent topology between most branches

in all trees, with robust root derived from CobN as an outgroup to BchH. Branch lengths are approximate, derived from the BchM phylogeny. B) Consensus phylogenies of steps in (bacterio)chlorophyll synthesis. Major radiations are collapsed at the phylum level. The Chloroflexi clade includes phototrophic Chloroflexia as well as multiple lineages of phototrophic Anaerolineae. The Proteobacteria clade includes Gemmatimonadetes and in some cases (BchL/N/B) the Synechococcus/Prochlorococcus clade of Cyanobacteria. Consensus tree for BchL, BchN, and BchB, subunits of the DPOR complex for conversion of protochlorophyllide a to chlorophyllide a, the last step shared in chlorophyll and bacteriochlorophyll synthesis pathways. Long branches between BchL/N/B and closest outgroups (BchX/Y/Z) resulted in poorly supported root placement, so tree is presented unrooted. The topology of the BchL/N/B tree is identical at the phylum level to the BchH/D/I/M tree, providing support for interpretation of a shared history of the entire "backbone" (bacterio)chlorophyll synthesis pathway and an inferred root for the BchL/N/B tree on the branch between the Cyanobacteria+Heliobacteria clade and the other phyla. Branch lengths are approximate, derived from the BchL phylogeny. C) Consensus phylogenies of steps in (bacterio)chlorophyll synthesis. Major radiations are collapsed at the phylum level. The Chloroflexi clade includes phototrophic Chloroflexia as well as multiple lineages of phototrophic Anaerolineae. The Proteobacteria clade includes Gemmatimonadetes and in some cases (BchL/N/B) the Synechococcus/Prochlorococcus clade of Cyanobacteria. Consensus tree for BchX, BchY, and BchZ, used for the conversion of chlorophyllide a to 3-vinyl-bacteriochlorophyllide a, the first dedicated step in the synthesis of bacteriochlorophylls a, b, and g, and therefore found in all characterized anoxygenic phototrophs. Long branches between BchX/Y/Z and closest outgroups (BchL/N/B) resulted in poorly supported root placement, so tree is presented unrooted. The topology of this tree is incongruent with those presented in A) and B), suggesting independent histories of HGT of bacteriochlorophyll-specific genes versus shared backbone (bacterio)chlorophyll synthesis genes. Branch lengths are approximate, derived from the BchX phylogeny.
(PDF)

**S3 Fig. Protein phylogeny with TBE supports for BchH.**
(JPG)

**S4 Fig. Protein phylogeny with TBE supports for BchD.**
(JPG)

**S5 Fig. Protein phylogeny with TBE supports for BchI.**
(PDF)

**S6 Fig. Protein phylogeny with TBE supports for BchM.**
(PDF)

**S7 Fig. Protein phylogeny with TBE supports for BchL.**
(PDF)

**S8 Fig. Protein phylogeny with TBE supports for BchN.**
(PDF)

**S9 Fig. Protein phylogeny with TBE supports for BchB.**
(PDF)

**S10 Fig. Protein phylogeny with TBE supports for BchX.**
(PDF)

**S11 Fig. Protein phylogeny with TBE supports for BchY.**
(PDF)

**S12 Fig. Protein phylogeny with TBE supports for BchZ.**
(PDF)

**S13 Fig. Protein phylogeny for BchN and BchB.**
(PDF)

**S14 Fig. Protein phylogeny for BchB and BchZ.**
(PDF)

**S15 Fig. Protein phylogeny for BchN and BchY.**
(PDF)

**S16 Fig. Protein phylogeny for BchY and BchZ.**
(PDF)

**S17 Fig.**
(PDF)

**S1 Table. Proteins involved in (bacterio)-chlorophyll synthesis from protoporphyrin IX.**
(DOCX)

**S2 Table.**
(TXT)

## Author Contributions

**Conceptualization:** Lewis M. Ward, Patrick M. Shih.

**Data curation:** Lewis M. Ward.

**Formal analysis:** Lewis M. Ward.

**Funding acquisition:** Lewis M. Ward.

**Investigation:** Lewis M. Ward, Patrick M. Shih.

**Methodology:** Lewis M. Ward.

**Project administration:** Lewis M. Ward, Patrick M. Shih.

**Resources:** Lewis M. Ward.

**Software:** Lewis M. Ward.

**Supervision:** Lewis M. Ward, Patrick M. Shih.

**Validation:** Lewis M. Ward, Patrick M. Shih.

**Visualization:** Lewis M. Ward, Patrick M. Shih.

**Writing – original draft:** Lewis M. Ward, Patrick M. Shih.

**Writing – review & editing:** Lewis M. Ward, Patrick M. Shih.

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
