## [Decision Letter · Decision Letter 0]

21 Oct 2020

PONE-D-20-27834

Granick Revisited: Synthesizing Evolutionary and Ecological Evidence for the Late Origin of Bacteriochlorophyll via Ghost Lineages and Horizontal Gene Transfer

PLOS ONE

Dear Dr. Ward,

Thank you for submitting your manuscript to PLOS ONE. After careful consideration, we feel that it has merit but does not fully meet PLOS ONE’s publication criteria as it currently stands. Therefore, we invite you to submit a revised version of the manuscript that addresses the points raised during the review process.

We look forward to receiving your revised manuscript.

Kind regards,

Chih-Horng Kuo, Ph.D.

Academic Editor

PLOS ONE

Journal Requirements:

3. Please note that in order to use the direct billing option the corresponding author must be affiliated with the chosen institute. Please either amend your manuscript or remove this option (via Edit Submission).

Reviewers' comments:

Reviewer's Responses to Questions

**Comments to the Author**

1. Is the manuscript technically sound, and do the data support the conclusions?

Reviewer #1: Yes

Reviewer #2: Yes

2. Has the statistical analysis been performed appropriately and rigorously? 

Reviewer #1: N/A

Reviewer #2: Yes

3. Have the authors made all data underlying the findings in their manuscript fully available?

Reviewer #1: Yes

Reviewer #2: Yes

4. Is the manuscript presented in an intelligible fashion and written in standard English?

Reviewer #1: Yes

Reviewer #2: Yes

5. Review Comments to the Author

Reviewer #1: The results in this article supports the Granick hypothesis that chlorophyll molecules are more ancient than bacteriochlorophyll molecules. The authors also mention the earliest phototrophs might perform anoxygenic photosynthesis using chlorophylls, and the later anoxygenic phototrophs use bacteriochlorophyll because of the competition of visible light with Cyanobacteria. The evolution process was achieved by horizontal gene transfer from ghost lineages. Those ideas are interesting and wroth for publishing, but there are several major and minor points that need to be addressed for clarity.

Major:

1) Figure 3: This figure is based on the results from Supplemental Figures 3-16. It is not easy for readers to read all the supplemental figures and draw the same conclusion as the authors did. A table with all the information included will be useful for readers to see the authors’ summarized results, such as the number of species/genes in the ghost lineages and how many of them are likely to be transferred to cyanobacteria or anoxygenic phototrophs.

2) In Figure 4, this manuscript discusses mainly on cyanobacteria and anoxygenic bacteria, and the authors performed phylogenetic analyses on them. It looks inappropriate to include algae and land plants in Figure 4, unless the authors provide analysis on genes in algae and land plants as well. Alternatively, the authors can provide supporting references on the evolution timeline of algae and land plants.

3) The conclusion is not concise and needs to be shortened.

Minor:

1) In Figure 1, the authors should mention full names of WPS2, RC1, RC2, PSI, and PSII before using those abbreviations.

2) The “Methods” section is suggested to be divided into several subsections for clarity.

3) There is a missing reference number 8.

Reviewer #2: The manuscript of Ward and Shih presents a fascinating hypothesis on the evolution of (bacterio)chlorophyll biosynthesis and photosynthesis, with some novel ideas about the direction and timing of certain gene transfer events. It is concise and well written, although perhaps a bit tough for a non-initiate of phylogenetics to follow. The paper will receive much interest from the community and stimulate further discussion on the evolution photosynthesis. I therefore recommend this work for publication.

I have only very minor comments to raise, detailed below:

Table 1. Most phototrophic proteobacteria display some aerobic metabolism or tolerance to oxygen, but there are examples of obligate anaerobes, e.g. Rhabdochromatium marinum

Pg 14. Anaerolineae lack BchLNB – since this statement is based on metagenomic data, could it be that these organisms are no longer phototrophic, having lost bchLNB? Alternatively, there are purple bacteria that lack bchLNB and rely on a light-activated POR acquired from cyanobacteria, could the same be true here? Ward et al, 2018a states that CP2_42A and JP3_7 appear to lack bchM and bchE as well as bchLNB - if bchM and bchE are absent, as they also appear to be in Ca. Thermochlorobacter aerophilum, a single enzyme may have evolved to replace both of these enzymes (activity of ChlM/BchM is essential for activity of AcsF/BchE). The genomes of diatoms also lack chlE/acsF, so these data suggest the presence of a third, or possibly fourth cyclase enzyme…

Cheng et al document that NifH can substitute for ChlL, rather than for ChlN.

Supplemental Fig 1. BciB is a second vinyl reductase, and there is evidence that this step occurs after C17=C18 reduction by POR in vivo. ChlP/BchP should be added alongside ChlG/BchG, tail reduction probably occurs after esterification.

6. PLOS authors have the option to publish the peer review history of their article (what does this mean?). If published, this will include your full peer review and any attached files.

Reviewer #1: No

Reviewer #2: No

---

## [Author Response · Author response to Decision Letter 0]

2 Dec 2020

Thank you for the helpful comments. We have revised our manuscript following these suggestions. We have attached our revised manuscript as well as a version marked up using the track changes function of Microsoft Word. Specific responses to reviewer comments are included below. 

Reviewer #1: The results in this article supports the Granick hypothesis that chlorophyll molecules are more ancient than bacteriochlorophyll molecules. The authors also mention the earliest phototrophs might perform anoxygenic photosynthesis using chlorophylls, and the later anoxygenic phototrophs use bacteriochlorophyll because of the competition of visible light with Cyanobacteria. The evolution process was achieved by horizontal gene transfer from ghost lineages. Those ideas are interesting and wroth for publishing, but there are several major and minor points that need to be addressed for clarity.

Major:

1) Figure 3: This figure is based on the results from Supplemental Figures 3-16. It is not easy for readers to read all the supplemental figures and draw the same conclusion as the authors did. A table with all the information included will be useful for readers to see the authors’ summarized results, such as the number of species/genes in the ghost lineages and how many of them are likely to be transferred to cyanobacteria or anoxygenic phototrophs.

Response: Supplemental Figures 3-16 provide the full dataset used to construct Figure 3, but these data are previously summarized in Figure 2 to help readers more easily visualize differences in phylogenetic relationships between photosynthesis proteins. We have now added Table 2 as a companion to Figure 3, summarizing our proposed set of HGT events. 

2) In Figure 4, this manuscript discusses mainly on cyanobacteria and anoxygenic bacteria, and the authors performed phylogenetic analyses on them. It looks inappropriate to include algae and land plants in Figure 4, unless the authors provide analysis on genes in algae and land plants as well. Alternatively, the authors can provide supporting references on the evolution timeline of algae and land plants.

Response: This is a good point. We have added a few lines summarizing the large body of work describing the derivation of eukaryotic photosynthesis from endosymbiosis of oxygenic cyanobacteria: “Eventually, the plastid endosymbiosis event would give rise to a wide diversity of eukaryotic oxygenic phototrophs (Reyes-Prieto et al, 2006; Archibald 2015; Shih et al, 2013); thus, the niche adaptation and competition with anoxygenic phototrophs would later be expanded beyond Cyanobacteria to include algae, and ultimately the rise of plants. Although the exact timing of the plastid endosymbiosis is still widely debated, the majority of studies agree that it is of Proterozoic origin, after the radiation of Cyanobacteria (Yoon et al, 2004; Shih and Matzke 2013; Gibson et al, 2018). Compared to anoxygenic phototrophic lineages, the chlorophyll requirements/composition of Cyanobacteria, algae, and plants are much more similar (i.e., Chl a and b) – albeit with some notable variations (Ho et al, 2016; La Roche et al, 1996; Gan et al 2014) – which is reflective of the common evolutionary history of these three major groups. “

3) The conclusion is not concise and needs to be shortened.

Response: We have moved a large portion of our conclusions section to a new “Combined perspectives on the evolution of photosynthesis” section. The conclusion is now much shorter and focuses more on big-picture implications. 

Minor:

1) In Figure 1, the authors should mention full names of WPS2, RC1, RC2, PSI, and PSII before using those abbreviations.

Response: This is a good point. We have added a key explaining these abbreviations in the figure caption. 

2) The “Methods” section is suggested to be divided into several subsections for clarity.

Response: Done. 

3) There is a missing reference number 8.

Response: Fixed.

Reviewer #2: The manuscript of Ward and Shih presents a fascinating hypothesis on the evolution of (bacterio)chlorophyll biosynthesis and photosynthesis, with some novel ideas about the direction and timing of certain gene transfer events. It is concise and well written, although perhaps a bit tough for a non-initiate of phylogenetics to follow. The paper will receive much interest from the community and stimulate further discussion on the evolution photosynthesis. I therefore recommend this work for publication.

I have only very minor comments to raise, detailed below:

Table 1. Most phototrophic proteobacteria display some aerobic metabolism or tolerance to oxygen, but there are examples of obligate anaerobes, e.g. Rhabdochromatium marinum

Response: Fair point. We have added a parenthetical note that there are some exceptions.

Pg 14. Anaerolineae lack BchLNB – since this statement is based on metagenomic data, could it be that these organisms are no longer phototrophic, having lost bchLNB? Alternatively, there are purple bacteria that lack bchLNB and rely on a light-activated POR acquired from cyanobacteria, could the same be true here? Ward et al, 2018a states that CP2_42A and JP3_7 appear to lack bchM and bchE as well as bchLNB - if bchM and bchE are absent, as they also appear to be in Ca. Thermochlorobacter aerophilum, a single enzyme may have evolved to replace both of these enzymes (activity of ChlM/BchM is essential for activity of AcsF/BchE). The genomes of diatoms also lack chlE/acsF, so these data suggest the presence of a third, or possibly fourth cyclase enzyme…

Response: We absolutely agree that there could be unidentified enzymes substituting in for BchE and BchM, but we’ve omitted discussion of these since it felt less essential to main story of the manuscript. We have revised the discussion here to clarify that there is evidence for production of bchl a in members of Anaerolineae despite the absence of both BchLNB and the POR complex: “Some phototrophic Chloroflexi in the Anaerolineae class lack BchLNB as well as genes for the alternative light-activated POR complex (Klatt et al. 2011, Ward et al. 2018a, Ward et al. 2020) despite evidence for fluorescence microscopy-based evidence for functional bacteriochlorophyll a synthesis in at least some of these organisms (Tank et al. 2017). This appears to be a derived trait based on their placement in BchHDI and BchXYZ trees. “ 

Cheng et al document that NifH can substitute for ChlL, rather than for ChlN.

Response: Corrected 

Supplemental Fig 1. BciB is a second vinyl reductase, and there is evidence that this step occurs after C17=C18 reduction by POR in vivo. ChlP/BchP should be added alongside ChlG/BchG, tail reduction probably occurs after esterification.

Response: Added BciB, ChlP, and BchG to the figure, along with a note in the caption stating that we have presented a simplified, linear view of the (b)chl biosynthesis process for the sake of clarity: “While some steps can be performed by multiple enzymes and some enzymes may act on multiple substrates, allowing some steps to be performed in different orders in different organisms or in parallel in a single organism, the (b)chl biosynthesis pathway is depicted here as a simplified linear pathway for the sake of clarity”

---

## [Decision Letter · Decision Letter 1]

30 Dec 2020

Granick Revisited: Synthesizing Evolutionary and Ecological Evidence for the Late Origin of Bacteriochlorophyll via Ghost Lineages and Horizontal Gene Transfer

PONE-D-20-27834R1

Dear Dr. Ward,

We’re pleased to inform you that your manuscript has been judged scientifically suitable for publication and will be formally accepted for publication once it meets all outstanding technical requirements.

Kind regards,

Chih-Horng Kuo, Ph.D.

Academic Editor

PLOS ONE

Additional Editor Comments (optional):

Reviewers' comments:

Reviewer's Responses to Questions

**Comments to the Author**

1. If the authors have adequately addressed your comments raised in a previous round of review and you feel that this manuscript is now acceptable for publication, you may indicate that here to bypass the “Comments to the Author” section, enter your conflict of interest statement in the “Confidential to Editor” section, and submit your "Accept" recommendation.

Reviewer #1: All comments have been addressed

Reviewer #2: (No Response)

2. Is the manuscript technically sound, and do the data support the conclusions?

Reviewer #1: Yes

Reviewer #2: Yes

3. Has the statistical analysis been performed appropriately and rigorously? 

Reviewer #1: Yes

Reviewer #2: Yes

4. Have the authors made all data underlying the findings in their manuscript fully available?

Reviewer #1: Yes

Reviewer #2: Yes

5. Is the manuscript presented in an intelligible fashion and written in standard English?

Reviewer #1: Yes

Reviewer #2: Yes

6. Review Comments to the Author

Reviewer #1: (No Response)

Reviewer #2: Supplemental Fig 1 linked in the manuscript file takes me to the original figure, which the authors should have updated.

Other than this I am satisfied with the resubmission.

7. PLOS authors have the option to publish the peer review history of their article (what does this mean?). If published, this will include your full peer review and any attached files.

Reviewer #1: No

Reviewer #2: No

---

## [Editor Report · Acceptance letter]

13 Jan 2021

PONE-D-20-27834R1 

Granick revisited: synthesizing evolutionary and ecological evidence for the late origin of bacteriochlorophyll via ghost lineages and horizontal gene transfer 

Dear Dr. Ward:

I'm pleased to inform you that your manuscript has been deemed suitable for publication in PLOS ONE. Congratulations! Your manuscript is now with our production department. 

Kind regards, 

on behalf of

Dr. Chih-Horng Kuo 

Academic Editor

PLOS ONE